Multi-modal affine fusion network for social media rumor detection

Fu Boyang
Sui Jie suijie@ucas.edu.cn
School of Engineering Science, University of Chinese Academy of Sciences , Beijing , China
Shanmuganathan Vimal
Electronic publication date: 2022 May 3
Publication date: 2022
Volume: 8
Electronic Location ID: e928
Received 2021 Nov 29; Accepted 2022 Mar 1
Copyright: © 2022 Fu and Sui
Copyright year: 2022
Copyright holder: Fu and Sui
License: This is an open access article distributed under the terms of the Creative Commons Attribution License, which permits unrestricted use, distribution, reproduction and adaptation in any medium and for any purpose provided that it is properly attributed. For attribution, the original author(s), title, publication source (PeerJ Computer Science) and either DOI or URL of the article must be cited.
License URL: https://creativecommons.org/licenses/by/4.0/

Keywords: Computer vision, Rumor detection, Social media fraud, Deep learning, Multimodality

Funding: The National Natural Science Foundation of China 61572459 This work is supported by The National Natural Science Foundation of China (Grant No. 61572459). The funders had no role in study design, data collection and analysis, decision to publish, or preparation of the manuscript.

==============================
With the rapid development of the Internet, people obtain much information from social media such as Twitter and Weibo every day. However, due to the complex structure of social media, many rumors with corresponding images are mixed in factual information to be widely spread, which misleads readers and exerts adverse effects on society. Automatically detecting social media rumors has become a challenge faced by contemporary society. To overcome this challenge, we proposed the multimodal affine fusion network (MAFN) combined with entity recognition, a new end-to-end framework that fuses multimodal features to detect rumors effectively. The MAFN mainly consists of four parts: the entity recognition enhanced textual feature extractor, the visual feature extractor, the multimodal affine fuser, and the rumor detector. The entity recognition enhanced textual feature extractor is responsible for extracting textual features that enhance semantics with entity recognition from posts. The visual feature extractor extracts visual features. The multimodal affine fuser extracts the three types of modal features and fuses them by the affine method. It cooperates with the rumor detector to learn the representations for rumor detection to produce reliable fusion detection. Extensive experiments were conducted on the MAFN based on real Weibo and Twitter multimodal datasets, which verified the effectiveness of the proposed multimodal fusion neural network in rumor detection.

Introduction

As Internet technology gradually matures, online social networking (OSN) has become the habitat in which social relationships are formed. Since OSN information is open and easily accessible, social networking software such as Weibo, Twitter, and Facebook have become the primary sources for millions of global users to receive news and information. They serve as essential approaches for Internet users to express their opinions. However, the authenticity of published information cannot be detected without supervision. Such social networking software has become the source of public opinion in hot events and news media.

For example, during the tenure of Barack Obama as the US President, a tweet from the “so-called” Associated Press said, “Two explosions occurred in the White House, and US President Barack Obama was injured.” Three minutes after the tweet was sent, the US stock index plunged like a “roller coaster,” and the market value of the US stock market evaporated by 200 billion US dollars within a short period, which had a tremendous effect on both the stock and bond futures. Soon after, the Associated Press issued a statement saying that its Twitter account had been hacked, and that tweet proved to be false news. Therefore, it is of great necessity to automatically detect social media rumors in the early stage, and this technology will be extensively applied with the rapid development of social networks.

Nowadays, online rumors are no longer in the single form of texts. Instead, they are often in multiple modalities that combine images and texts. Figure 1 shows the cases of rumors in the Twitter dataset, displaying the texts and images of each tweet. In Fig. 1A, the news is fake based on the images and texts; it is hard to identify whether the news in Fig. 1B is true or not, but the images are fake; we cannot determine the authenticity of the news in Fig. 1C based on the images, but we can confirm that the information is not true according to the texts.

Figure 1 Three forms of rumors on Weibo and Twitter datasets.

(A) Text: MH-370 has been found near Bermuda; (B) Text: Sharks in the street…; (C) Text: Sandy! Is this true or just some ‘trick or treat’ joke? Image credit: Boididou et al. (2015).

Currently, most methods used to detect social media rumors automatically are based on traditional machine learning (Tacchini et al., 2017; Dongo et al., 2020; Choi et al., 2020; Chou, Liu & Lee, 2021) and deep learning (Song et al., 2021; Jinshuo et al., 2020; Rani, Das & Bhardwaj, 2021; Gokhale et al., 2020). The neural network (Rauf, Bangyal & Lali, 2021), and other learning mechanisms such as federated learning (Gao et al., 2021) can learn the constantly changing high-dimensional feature representation of posts in the training process with the superior ability to extract features. The currently available research on rumor detection primarily focus on single modality (Jin, Wu & Guo, 2020; Abdulrahman & Baykara, 2020; Luo et al., 2021; Balpande et al., 2021), while multi-modal researches are still in infancy, and only a few recent researches have tried to explore the multiple modalities (Jin et al., 2017; Wang et al., 2018; Khattar et al., 2019; Jinshuo et al., 2020; Huang et al., 2019).

In current studies, the features of images and texts are mostly fused through feature concentration and averaging results. Nevertheless, this single fusion method fails to represent the posts fully. First, it cannot solve the problem caused by the difference in semantic correlation between texts and images in rumors and non-rumors; second, the semantic gap cannot be overcome. Moreover, unlike paragraphs or documents, the texts in posts that are usually short fail to provide enough context information, making our classification fuzzier and more random.

This paper introduces a new end-to-end framework to solve the above problems. This framework is known as the multi-modal affine fusion network (MAFN). In the proposed model, employing affine fusion, we fused the features of images and texts to reduce the semantic gap and better capture the semantic correlation between images and texts. Entity recognition was introduced to improve the semantic understanding of texts and enhance the ability of rumor detection models. MAFN can gain multi-modal knowledge representation by processing posts on social media to detect rumors effectively. This paper makes the following three contributions: We proposed the multi-modal affine fusion network (MAFN) combined with entity recognition for the first time better to capture the semantic correlation between images and texts.

The proposed MAFN model enriched the semantic information of text with entity recognition, and entity recognition was fused with the extracted textual features to improve the semantic comprehension of text.

Experiments show that the MAFN model proposed in this paper can effectively identify rumors on Weibo and Twitter datasets and is superior to currently available multi-modal rumor detection models.

Related work

In early research on rumor detection (Castillo, Mendoza & Poblete, 2011; Kwon et al., 2013), the rumor detection model was mainly established based on the differences between the features of rumors and factual information. Castillo, Mendoza & Poblete (2011) designed a simple model to evaluate the authenticity of information on Twitter by counting the frequency of words, punctuation marks, expressions, and hyperlinks in texts. On this basis, Kwon et al. (2013) used the communication structure to build rumors into a communication network and put forward 15 structural features, including the mid-values of network depth and width. Yang et al. (2012) introduced other client-based and location-based functions to identify rumors on Sina Weibo. However, it is time and energy-consuming to design these features manually, and the language patterns are highly dependent on specific time and knowledge in corresponding fields. Therefore, these features cannot be correctly understood.

Rumors on social media have gradually transformed from text-based to multi-modal rumors that combine both texts and images. Data in different modalities can complement each other. An increasing number of researchers have tried to integrate visual information into rumor detection. Singh, Ghosh & Sonagara (2021) manually designed textual, and image features in four dimensions, i.e., content, organization, emotions, and manipulation, and eventually fused multiple features to detect rumors. Jin et al. (2017) detected rumors by fusing the image and textural features of posts using the RNN combined with the attention mechanism. However, multi-modal features still depend highly on specific events in the dataset, which will weaken the model’s generalization ability. Therefore, Wang et al. (2018) put forward the EANN model that connected the visual features and textual features of posts in series and applied the event discriminator to remove specific features of events and learn the shared features of rumor events. Experiments show that this method can detect many events that are difficult to distinguish in a single modality.

Ma et al. (2016) introduced recurrent neural networks (RNN) to learn hidden representations from the texts of related posts and used LSTM, GRU, and 2-layer GRU to model text sequences, respectively. It was the first attempt to introduce a deep neural network into post-based rumor detection and achieve considerable performance on real datasets, verifying the effectiveness of deep learning-based rumor detection. Yu et al. (2017) used a convolutional neural network (CNN) to obtain critical features and their advanced interactions from the text content of related posts. Nonetheless, CNN is unable to capture long-distance features. Hence, Chen et al. (2019) applied an attention mechanism to the detection of network rumor and proposed a neural network model with deep attention. This model extracts adequate information and essential features from highly repeated texts, which solves the problems of excessive redundancy of texts in the data to be tested and weak information links between remote sites.

According to Khattar et al. (2019), a single fusion method cannot effectively represent the posts. So, they used the encoder and decoder to extract the features of images and texts and learned across modalities with the help of Gaussian distribution. Jinshuo et al. (2020) put the text vector, the text vector in the image, and the image vector together, and then processed them using Gaussian distribution to get a new fusion vector to discover the association between the two modalities of hidden representation. Besides learning the text representation of posts, Zhang et al. (2019) retrieved external knowledge to supplement the semantic representation of short posts and used conceptual knowledge as additional evidence to improve the performance of the rumor detection model.

Methodology

This paper introduced the four modules of the proposed MAFN model in this Section, i.e., the entity recognition enhanced textual feature extractor, the visual feature extractor, the multi-modal affine fuser, and the rumor classifier. Furthermore, we described the integration of the proposed modules to represent and detect rumors.

We instantiated tweets on Weibo and Twitter. The total tweets were expressed as S={t1,t2,…,tn}, and each tweet was expressed as t = {T,E,V}, where T denotes the text content of the tweets, E represents the entity content extracted from the tweets, and V stands for the visual content matched with the tweets. L={L1,L2,…,Lm} denotes the corresponding rumor and non-rumor tags of tweets. This paper aims to learn a multi-modal fusion classification model F by using the total tweets S and the corresponding tag sets L. F can predict rumors on unmarked social media. Figure 2 shows the framework of the proposed model.

Figure 2 The model diagram of the proposed multimodal network MAFN.

The yellow part represents the visual feature extractor, the blue part denotes the entity recognition enhanced textual feature extractor, the pink part stands for the multi-modal affine fuser, and the green part refers to the rumor detector.

The entity recognition enhanced textual feature extractor and obtained the joint representation Ru of text using Bert pre-training and self-attention mechanism. The visual feature extractor used the pre-trained model VGG19 to capture visual semantic feature Rv. The multi-modal affine fuser fused the joint representation and visual representation to obtain Rs, and the rumor classifier was utilized in the end to detect rumors.

Entity recognition enhanced textual feature extractor

Extraction of text representation

Text representation is a short text representation generated from tweets. Our model extracted the feature vector of tweets through the Bert model to better capture the context’s possible meaning and semantic meaning. Bert is a natural language processing model with the transformer bidirectional encoder representation as to the core, which can better extract the text context representation bidirectionally. By inputting the sequential vocabulary of the words in the tweets, the words were first embedded into the vector. The dimension of the ith word in the sentence is denoted by m, which is expressed as Wi∈ Rm, and by inputting it into the sentence, S, it can be expressed as:

(1) S=[W0,W1,W2,…,Wp]

where, S∈ Rm*p, p denotes the total number of words, W0 denotes [CLS], and Wp represents [SEP]. By inputting the complete texts of tweets into the Bert model, we obtained the feature vector of the given sentence as

Sf=[Wf0,Wf1,Wf2,…,Wfp]

Then the sentence feature vectors Sfn were given to the two fully connected layers. The above steps can be defined as follows:

(2) Rt′=σ(Wft2⋅σ(Wft1⋅Sf+bt1)+bt2)

where W ft1 denotes the weight matrix of the first fully connected layer with activation function, W ft2 represents the weight matrix of the second fully connected layer with activation function, and bt1 and bt2 are the bias terms.

The attention-based neural network can better obtain relatively long dependencies in sentences. The self-attention mechanism is a kind of attention mechanism that associates different positions of a single sequence to calculate the representation of the same sequence. To enable the model to learn the correlation between the current word and the other parts of the sentence, we added the self-attention mechanism after the fully connected layer, the process of which was expressed as follows:

(3) Attself=softmax[QT⋅KT⊤/m]⋅VT

where, QT = Rt′ × WQT, KT = Rt′ × WKT, VT = Rt′ × WVT · WQT, WKT, WVT denote the three different matrices learned by Q, K, and V, respectively. To make the model automatically recognize the importance of each word, degrade unimportant features to their original features, and process essential features using the self-attention mechanism, we used the residual connection to extract the features better. Figure 3 shows the architecture of a residual self-attention. A building block was defined as:

Figure 3 The architecture of a residual self-attention.

(4) Rt=Attself+Rt′

where, Rt denotes the eventually extracted text representation, Rt ∈ Rk.

Extraction of entity representation

Named entity recognition identifies person names, place names, and organization names in a corpus. It was assumed that the combination of entity tagging and text coding in a post could supplement the semantic representation of the short text of the post in a certain way so that the model could identify rumors and non-rumors more accurately. Explosion AI developed spacy, a team of computer scientists and computational linguists in Berlin, and its named entity recognition model was pre-trained on OntoNotes 5, a sizeable authoritative corpus. In this paper, Spacy was applied to train the two datasets and extract the entities of posts. There were 18 kinds of identifiable entities.

First of all, we identified the recognizable word Wi as the entity e ∈ Es in every sentence S=[W0,W1,W2,…,Wp] of the tweet, and then obtained the tag L∈{L1,L2,…,Ln} corresponding to this entity, where Li is one of the tags {PERSON, LANGUAGE,…,LOC}. For instance, to instantiate a piece of text, we instantiated the entities in the text, as shown in Fig. 4. The extracted entity LEuropean = {NORP}, NORP means nationalities or religions or political groups; LGoogle = {ORG}, OPG represents companies, agencies, institutions etc.

Figure 4 Illustration of entity refining process.

Based on the obtained Li, the corresponding entity tags were connected in series to capture semantic features by Bert. Ef ∈ Rk, where Rk denotes the embedding dimension of tags. By inputting Ef into the residual attention mechanism, we gained Re ∈ Rk.

In the end, we combined the extracted text representation with the entity representation to obtain the joint representation Ru, Ru ∈ Rk, which was defined as follows:

(5) Ru=add Re,Rt

Visual feature extractor

Images in tweets form the input into the visual feature extractor. This proposed framework used the pre-trained model VGG-19 and added two fully connected layers in the last layer to more comprehensively extract the visual features matched with the rumors in the tweet. According to the parameters unchanged after pre-training, VGG-19 adjusted the representation dimension of final visual features to k through two fully connected layers. We added the batch normalization layer and drop-out layer between the two fully connected layers and the activation function to prevent overfitting during the extraction of image representation. The eventually obtained feature of visual representation was expressed as Rv, where Rv ∈ Rk. The equation for extracting image features was defined as follows:

(6) Rv′=Wfv2⋅σ(BN(Wfv1⋅Rvgg+bv1))+bv2

(7) Rv=Dropout(σ(BN(Rv′)))

where, Rvgg represents the visual features extracted from the network in the pre-trained model VGG19, σ is the activation function, W fv1 denotes the weight matrix of the first fully connected layer with the activation function, and bv1 and bv2 are the bias terms.

Multi-modal affine fuser

Affine transformation transforms into another vector space via linear transformation and translation. Through affine transformation, the multi-modal affine fuser fuses the multi-modal features extracted by the entity recognition enhanced textual feature extractor and the visual feature extractor, the joint representation and visual features of text and entity. It was assumed that the data of the two modalities could be fused more closely and the high-level semantic correlation could be better extracted. The corresponding equation was defined as follows:

(8) Rc=FRv⋅Ru+H(Rv)

where, Rc is the feature Rc ∈ Rk gained after the fusion of all features, and F⋅ and H⋅ were fitted by the neural network. After extracting the fused features, in order to get more robust features, we reconnected the fused features with the textual features to obtain the total feature Rs. The equation was expressed as:

(9) Rs=Rc⊕Rt

where, ⊕ denotes concatenation.

Rumor detector

The rumor detector, based on the multi-modal affine fuser, sent the finally obtained multi-modal feature Rs to the multilayer perceptron for classification to judge whether the message was a rumor or not. The rumor detector consists of multiple completely connected layers with softmax. The rumor detector was expressed as G(Ris, θ), where θ represents all the parameters in the rumor detector, and Ris denotes the multi-modal representation of the case of the ith tweet. The rumor detector was defined as follows:

(10) pi=G(Rsi,θ)

where pi denotes the probability that the ith post input by the detector is a rumor, in the process of model training, we selected the cross-entropy function as the loss function, which was expressed as follows:

(11) Loss=∑i=1N−[Li×log(pi)+(1−Li)×log(1−pi)]

where, Li denotes the tag of the tweet in the i-th group, and N refers to the total number of training samples.

Experiments

This section first described the datasets used in the experiment, namely two social media datasets extracted from the real world. Secondly, we briefly compared the results obtained by the most advanced rumor detection method and those gained by the model proposed in this paper. Through the MAFN ablation experiment, we compared the performances of different models.

Datasets

To fairly evaluate the performance of the proposed model, we used two standard datasets extracted from the real world to assess the rumor detection framework of the MAFN. These two datasets were composed of rumors and non-rumors collected from Twitter and Weibo, which simulated the natural open environment to some extent. They are currently the only datasets with paired image and text information.

Weibo dataset

The Weibo dataset is a dataset proposed by Jin et al. (2017) for rumor detection. It consists of the data collected by Xinhua News Agency, an authoritative news source in China, and the website of Sina Weibo and the data verified by the official rumor refuting system of Weibo. We preprocessed the dataset using a method similar to that put forward by Jin. First, locality sensitive hashing (LSH) was applied to filter out the same images and then delete irregular images such as very small or very long images to ensure that images in the dataset were of uniform quality. In the last step, the dataset was divided into the training and test sets. The ratio of tweets in training set to those in the test set was 8:2.

Twitter dataset

The Twitter dataset (Boididou et al., 2015) was released to verify the task of social media rumor detection. This dataset contains about 15,000 tweets focusing on 52 different events, and each tweet is composed of texts, images, and videos. The ratio of concentrated development set to test set in the dataset is 15:2, with the ratio of rumors to non-rumors being 3:2. Since this paper mainly studies the fusion of texts and images, we filtered out all tweets with videos. The ratio of development set and test set used to train the proposed model is the same as above.

Experiment setting

The feature dimension of the images processed by VGG19 was 1,000; the image features were extracted and embedded by two linear layers to obtain the feature dimension. After applying Bert and the linear layer were processed, the texts and entities were turned into 32-dimensional vectors. The entire training epochs was 50, and the batch size was 32. Adam served as the model optimizer during the training of the model. The initial learning rate was 0.001, and then lr varied with epoch based on the following equation:

(12) p=float(epoch)/100

(13) lr=0.001/(1.+10∗p)∗∗0.75

Baselines

To verify the performance of the proposed multi-modal rumor detection framework based on knowledge attention fusion, we compared it with the single-modal methods, i.e., Textual and Visual, and five new multi-modal models. Textual and Visual were the subnetworks of the MAFN. The following are relatively new rumor detection methods for the comparative analysis: Neural Talk generates the words that describe images using the potential representations output by the RNN. Using the same structure, we applied the RNN to output the joint representation of images and texts in each step and then fed the representation into the fully connected layer for rumor detection and classification.

EANN (Wang et al., 2018): extracted textual features using Text-CNN, processes image features with VGG19 and then splices the two types of features together. With the features of specific events removed by the event discriminator, the remaining features were input into the fake news detector for classification.

MVAE (Khattar et al., 2019): used the structure of encoder-decoder to extract the image and textual features and conducted cross-modal learning with Gaussian distribution.

att-RNN (Jin et al., 2017) uses the RNN combined with the attention mechanism to fuse three modalities, i.e., image, textual, and user features. For a fair comparison, we removed the feature fusion in the user feature part of att-RNN, with the parameters of other parts being the same as those of the original model.

MSRD (Jinshuo et al., 2020) obtains a new fusion vector for classification by splicing textual features, textual features in images, and visual features extracted by VGG19 using Gaussian distribution.

VQA is applied in the field of visual questioning and answering. Initially a multi-classification task, the image question-and-answer task was changed to a binary classification task. We used a single-layer LSTM with 32 hidden units to detect and classify rumors.

Performance comparison

Table 1 shows the baseline results of single-modal and multi-modal models as well as the performances of the MAFN on two datasets in terms of the accuracy, precision, recall, and F1 of our rumor detection framework. MAFN performed better than the baseline models. The single textual model outperformed the single visual model on the Twitter dataset. Although the image features learned by visual features with the help of VGG-19 had better performance in rumor detection, the extraction of textural features was improved by Bert pre-training and residual attention. However, the single-modal model performed much. Among currently available multi-modal models, att-RNN uses LSTM and attention mechanism to process text representation, but it is not as good as EANN, which shows that EANN’s event discriminator can better improve the model when it comes to rumor detection. The variational autoencoder proposed by MVAE can better discover multi-modal correlation, and it outperforms EANN. MAFN outperformed all baselines in terms of accuracy, precision, and F1, with high accuracy increasing from 82.7% to 84.2% and the F1 score going up from 82.9% to 84.0%. This verifies the effectiveness of MAFN in rumor detection.

Table 1 Comparison of performances of MAFN and other methods on Twitter and Weibo datasets.

Dataset	Method	Accuracy	Precision	Recall	F1	
Twitter	Textual	0.551	0.680	0.605	0.520	
	Visual	0.512	0.655	0.59	0.505	
	NeuralTalk	0.610	0.728	0.504	0.595	
	VQA	0.631	0.765	0.509	0.611	
	att-RNN	0.664	0.749	0.615	0.676	
	MSRD	0.685	0.725	0.636	0.678	
	EANN	0.715	0.822	0.638	0.719	
	MVAE	0.745	0.801	0.719	0.758	
	MAFN	0.771	0.790	0.782	0.787	
Weibo	Textual	0.774	0.679	0.812	0.739	
	Visual	0.633	0.523	0.637	0.575	
	NeuralTalk	0.717	0.683	0.843	0.754	
	VQA	0.773	0.780	0.782	0.781	
	att-RNN	0.779	0.778	0.799	0.789	
	MSRD	0.794	0.854	0.716	0.779	
	MVAE	0.824	0.854	0.769	0.809	
	EANN	0.827	0.847	0.812	0.829	
	MAFN	0.842	0.861	0.821	0.840	

A similar trend was found on the Weibo dataset. The textual model is superior to the visual model among the single-modal models. The accuracy of single text reaches 77.4%, which verifies the effectiveness of Bert pre-training and residual self-attention mechanism in improving semantic representation. Among the multi-modal methods, att-RNN, EANN, and MSRD proposed for this task outperform NeuralTalk and VQA, proving the necessity of improving modal fusion. The proposed MAFN achieved the best performance among other state-of-the-art models, with accuracy increasing from 74.5% to 77.1% and the F1 score rising from 75.8% to 78.7%. This implies that the proposed model can better extract the multi-modal joint representation of images and texts.

Component analysis

To further analyze the performance of each part of the proposed model and to better describe the necessity of adding entity recognition and affine model, we carried out corresponding ablation experiments. We designed several comparison baselines, including simplified single-modal and multi-modal variants that removed some original models’ components. The Weibo dataset contains a greater variety of events without strong specificity, better reflecting the rumors in the real world. Therefore, we ran the newly designed simplified variants on the Weibo dataset.

As shown in Table 2, “w/o -entity” denotes the proposed MAFN without entity recognition module; “w/o -affine fusion” means removing affine fusion but retaining texts for entity recognition. Images and entity recognition were directly connected in series with the joint representation of texts. “w/o entity+ affine fusion” removed both entity and affine modules. “Text-only” refers to the single-text experiment. After pre-training the text using Bert, we connected the texts to the two fully connected layers and then accessed the residual self-attention to detect rumors directly. We conducted it for comparison. “Entity-Link-Only” results from rumor text detection carried out by only model branch entities. “w/o image” refers to the experiment without images, but only the combination of texts and entities. Furthermore, Table 2 indicates the performance of the simplified variant of MAFN. The experimental results show the necessity for the model to use affine fusion and enhance entity recognition. With entity-link added, the accuracy of single-modal text classification was increased from 77.4% to 79.9%, and F1 increased from 73.9% to 77.2%. A total of 1.9% also improved the accuracy of image text fusion due to the introduction of entity branches. It was found that entity branches could supplement semantic representation, proving our idea effective. According to the experimental results, if we remove affine fusion, the accuracy of MAFN will decrease by 1.3%, and F1 will also decline by 2.4%. If images and texts are only connected without adding fusion and supplement, the accuracy will be lower. This proves the effectiveness of MAFN in rumor detection. MAFN can achieve more reliable multi-modal representation.

Table 2 Variants of the proposed MAFN’s performance on Weibo datasets.

Method	Accuracy	Precision	Recall	F1	
MAFN	0.842	0.861	0.821	0.840	
w/o entity	0.836	0.826	0.826	0.826	
w/o affine fusion	0.829	0.800	0.832	0.816	
w/o entity+ affine fusion	0.819	0.750	0.852	0.797	
Text-only	0.774	0.679	0.812	0.739	
Entity-Link-Only	0.549	0.429	0.529	0.474	
w/o image	0.799	0.719	0.834	0.772	

Case study performance visualization

A qualitative analysis was performed on MAFN. After analyzing and ranking the examples of rumors successfully classified by MAFN, we selected the best two examples on Twitter and Weibo and showed them in Figs. 5 and 6, respectively. Without the support of affine fusion and entity recognition, the examples in Twitter could not be detected. Since the model failed to effectively capture the relationship between texts and images, these examples were misjudged as non-rumors. Insufficient text information and the absence of close connections between information and images are the reasons why the examples in Weibo could not be detected using “w/o entity+ affine fusion.” However, we can identify rumors with affine fusion by judging the image features.

Figure 5 Examples of successfully detecting rumors on Twitter by MAFN.

Image credit: Boididou et al. (2015).

Figure 6 Examples of successfully detecting rumors on Weibo by MAFN.

Image credit: Jin et al. (2017).

Conclusion

This paper proposed an affine fusion network combined with entity recognition. This network accurately identifies rumors using the affine fusion between the entity recognition joint representation of images and texts. When extracting text representation, we used Bert to generate sentence vector features and learn semantics by extracting knowledge from the outside through entity recognition. Moreover, affine fusion was used for multi-modal fusion to better summarize the invariant features of new events. The Twitter and Weibo datasets experiments show that the proposed model is robust and performs better than the most advanced baselines. In the future, we plan to capture and identify rumor propagation in the field of rumor text and short videos to strengthen the generalization ability of the multi-modal fusion model.

Additional Information and Declarations

Competing Interests

Author Contributions

Data Availability

The authors declare that they have no competing interests.

Boyang Fu conceived and designed the experiments, performed the experiments, analyzed the data, performed the computation work, prepared figures and/or tables, and approved the final draft.

Jie Sui conceived and designed the experiments, performed the experiments, analyzed the data, authored or reviewed drafts of the paper, and approved the final draft.

The following information was supplied regarding data availability:

The Weibo dataset is available at GitHub: https://github.com/wangyajun-ops/Weibo-dataset.

The Twitter dataset is available at GitHub: https://github.com/MKLab-ITI/image-verification-corpus.

The code is available at GitHub: https://github.com/Nopeeeee1122/multimodal-

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
