# Peer review of "Multi-modal affine fusion network for social media rumor detection"

_PeerJ Computer Science, doi:10.7717/peerj-cs.928_

## Round 0.1 · original submission · Major Revisions

Authors are asked to pay more attention in results section, and address each of the reviewers' comments.

Reviewer 1 ·

Basic reporting

Summary

- In this study, the authors proposed the multimodal affine fusion network (MAFN) combined with entity recognition, a new end-to-end framework that fuses multimodal features to detect rumors effectively. The MAFN mainly consists of four parts: the entity recognition enhanced textual feature extractor, the visual feature extractor, the multimodal affine fuser, and the rumor detector. The entity recognition enhanced textual feature extractor is responsible for extracting textual features that enhance semantics with entity recognition from posts. The visual feature extractor extracts visual features. The multimodal affine fuser extracts the three types of modal features and fuses them by the affine method, and it cooperates with the rumor detector to learn the representations for rumor detection to produce reliable fusion detection. Extensive experiments were conducted on the MAFN based on real Weibo and Twitter multimodal datasets, which verified the effectiveness of the proposed multimodal fusion neural network in rumor detection.
2. Strength

- this paper firstly introduced the four modules of the proposed MAFN model, i.e., the entity recognition enhanced textual feature extractor, the visual feature extractor, the multi-modal affine fuser, and the rumor classifier. Secondly, this paper describes how to integrate the four modules to represent and detect rumors.
3. Weakness
- Lack of novelty of research. textual feature extractor -based problem solving is a very common approach in the recent deep learning field, and post-processing is also difficult to consider as a new algorithm.
- The part about the learning scenario is confusing. A more understandable explanation is needed for training and testing.
- The entity resolution feature is quite confusing, and difficult to understand.
4. Minor comments
- There is an error in the reference. I haven't looked at all of them in detail.
- The manuscript is not well organized. The introduction section must introduce the status and motivation of this work and summarize with a paragraph about this paper.
- What are the limitations of the related works
-Are there any limitations of this carried out study?
-How to select and optimize the user-defined parameters in the proposed model?

Experimental design

please see above

Validity of the findings

please see above

Additional comments

please see above

·

Basic reporting

The authors proposed the multimodal affine fusion network (MAFN) combined with entity recognition, a new end-to-end framework that fuses multimodal features to detect rumors effectively.

Experimental design

The MAFN mainly consists of four parts: the entity recognition enhanced textual feature extractor, the visual feature extractor, the multimodal affine fuser, and the rumor detector. The entity recognition enhanced textual feature extractor is responsible for extracting textual features that enhance semantics with entity recognition from posts. The visual feature extractor extracts visual features. The multimodal affine fuser extracts the three types of modal features and fuses them by the affine method, and it cooperates with the rumor detector to learn the representations for rumor detection to produce reliable fusion detection.

Validity of the findings

The manuscript sounds technically good, I have the following concerns that should be addressed before any decision.

Grammatical mistakes
1. However, without supervision, the authenticity of published information cannot be detected--> should be ... "However, the authenticity of published information cannot be detected without supervision".
2. With the rapid development of the Internet, people obtain abundant information from social media such as Twitter and Weibo every day. However, due to the complex structure of social media, many rumors with corresponding images are mixed in real information to be widely spread, which misleads readers and exerts negative effects on society --> should be---> "With the rapid development of the Internet, people obtain much information from social media such as Twitter and Weibo every day. However, due to the complex structure of social media, many rumors with corresponding images are mixed in real information to be widely spread, which misleads readers and exerts adverse effects on society."
3. Established based on the multi-modal affine fuser, the rumor detector sent the finally obtained multi-modal feature--> "Based on the multi-modal affine fuser, the rumor detector sent the finally obtained multi-modal feature."
Minor Changes:
1. The author should provide only relevant information related to this paper and reserve more space for the proposed framework.
2. The theoretical perceptive of all the models used for comparison must be included in the literature.
3. What are the real-life use cases of the proposed model? The authors can add a theoretical discussion on the real-life usage of the proposed model.
4. However, the author should compare the proposed algorithm with other recent works or provide a discussion. Otherwise, it's hard for the reader to identify the novelty and contribution of this work.
5. The descriptions given in this proposed scheme are not sufficient that this manuscript only adopted a variety of existing methods to complete the experiment where there are no strong hypotheses and methodical theoretical arguments. Therefore, the reviewer considers that this paper needs more works.

Reviewer 3 ·

Basic reporting

The abstract should be reformulated. The abstract is an extremely important and powerful representation of the article. The authors have to clarify what is the novelty of this paper in abstract.
• Reduce challenges list as much as you can.
• Provide the related works clearly highlight the main gap.
• Authors have proposed three algorithms, but I do not understand which one is used in the comparison with state-of-the-art works.
• Figures 5abcd, have same label which quite confusing what is the point from each figure.
• Figures looks very fuzzy, and resolution of image is poor.
• Replace Case Study Performance Visualization with the discussion section as it is very poor and more deep discussion is needed for findings of the study.
• Conclusion needs to be improved. The most important obtained results should be briefly and clearly mentioned through the support of numerical data in the conclusion.
• The details in this manuscript are vague, especially in the depth feature extraction, which is the major defect of this paper. In addition, there is a gap between the experimental condition and the real scene. So this method can not be applied to the real scene effectively.
• There are many words and figures about the background/architectures of the proposed networks used in this paper that can be omitted. These proposed networks are in the field for a while and they are known by most likely every researcher in the field.
• There is a lack of comparison with other studies in the discussion. I do know that from the “related work” introduced in this paper that most previous study provides a very high accuracy/statistic in a much smaller dataset. The quantitative results are lower if just compare to the numeric values. However, the model in this study could be more robust than other previously published models applying your dataset using other models.

Experimental design

see above

Validity of the findings

see above

Additional comments

see above

---

## Round 0.2 · Minor Revisions

There are some minor things that should be considered before publication, like the organization of the paper. Please check that all tables and figures are cited appropriately in-text,

Reviewer 1 ·

Basic reporting

I'm satisfied with the current version.

Experimental design

I'm satisfied with the current version.

Validity of the findings

I'm satisfied with the current version.

·

Basic reporting

The paper seems improved as compared to previous versions. Hence, it is acceptable for publication.

Experimental design

The paper seems improved as compared to previous versions. Hence, it is acceptable for publication.

Validity of the findings

The paper seems improved as compared to previous versions. Hence, it is acceptable for publication.

Reviewer 3 ·

Basic reporting

Thanks, although the manuscript is improved, however, there are some minor things that should be considered before publication.

- For example, the organization of the paper. All tables and figures should be cited in order from low to high.

Experimental design

Looks good to me

Validity of the findings

Looks good to me

Additional comments

- Some grammatical and punctuation issues exist in the paper. It should be rectified.
- The figure quality still needs enhacments.

---

## Round 0.3 · accepted · Accept

All comments have been addressed.

---

## Author Rebuttal · Round 0.3

# Response Sheet

## Multi-modal affine fusion network for social media rumor detection

**Editor Comments**

There are some minor things that should be considered before publication, like the organization of the paper. Please check that all tables and figures are cited appropriately in-text.

**Response:** Thank you for allowing us resubmission, as per your suggestion we have updated the organization section and cross verified the table and figure citations. please see the revised version as track changes. Also, we covered all the reviewers' comments.

[# PeerJ Staff Note: On line 160, Figure 4 is written "Figure ??EQ4" #]

**Response:** Thank you for your concern, as per your suggestion we have corrected that error. Please see the revise version.

[# PeerJ Staff Note: The review process has identified that the English language must be improved. PeerJ can provide language editing services - please contact us at copyediting@peerj.com for pricing (be sure to provide your manuscript number and title) #].

**Response:** Thank you for your concern, as per your reviewer suggestions, we have proof read the article carefully and solved grammatical and sentence based mistakes, please see the updated version.

**Reviewer 3**

Thanks, although the manuscript is improved, however, there are some minor things that should be considered before publication.

- For example, the organization of the paper. All tables and figures should be cited in order from low to high.

**Response:** Thank you for your concern, as per your suggestion we have updated the organization section and cross verified the table and figure citations. please see the revised version as track changes.

Additional comments

- Some grammatical and punctuation issues exist in the paper. It should be rectified.

**Response:** Thank you for your concern, as per your reviewer suggestions, we have proof read the article carefully and solved grammatical and sentence based mistakes, please see the updated version.

- The figure quality still needs enhacments.

**Response:** Thank you for your concern, as per your suggestion, we have improved the quality of figures. Please see the updated version of manuscript.